# Neural FFTs for Universal Texture Image Synthesis

**Morteza Mardani**[*], **Guilin Liu**[*]**, Aysegul Dundar, Shiqiu Liu,**
**Andrew Tao, Bryan Catanzaro**
NVIDIA
{mmardani,guilinl,adundar,edliu,atao,bcatanzaro}@nvidia.com

## Abstract

Synthesizing larger texture images from a smaller exemplar is an important task in graphics and vision. The conventional CNNs, recently adopted for synthesis, require to train and test on the same set of images and fail to generalize to unseen images. This is mainly because those CNNs fully rely on convolutional and upsampling layers that operate locally and not suitable for a task as global as texture synthesis. In this work, inspired by the repetitive nature of texture patterns, we find that texture synthesis can be viewed as (local) *upsampling* in the Fast Fourier Transform (FFT) domain. However, FFT of natural images exhibits high dynamic range and lacks local correlations. Therefore, to train CNNs we design a framework to perform FFT upsampling in feature space using deformable convolutions. Such design allows our framework to generalize to unseen images, and synthesize textures in a single pass. Extensive evaluations confirm that our method achieves state-of-the-art performance both quantitatively and qualitatively.

## 1 Introduction

Texture synthesis is the expansion of a small texture example to an arbitrarily larger size while preserving the structural content. It is a challenging task given the wide range of textures a synthesizer should handle. Textures can have regular patterns such as brick walls, or, irregular patterns such as pebbles on the beach. A good synthesizer thus must be: 1) *universal* to synthesize a wide variety of textures patterns, 2) *fast* to synthesize in real-time for interactive tasks.

Motivated by the success of deep learning in image-to-image translation tasks, contemporary methods for texture synthesis rely on CNNs. Earlier works employ pre-trained CNNs as feature extractors to match the feature statistics of input and output via iterative optimization that is prohibitively slow [18, 35, 37]. To speed up synthesis, later works train *feed-forward* CNNs to learn the end-to-end synthesis map that expands textures in a single pass [37, 66, 52, 36].

Those CNNs typically require to train and test on the same set of images, that cannot generalize to unseen textures. We hypothesize that the poor generalization is attributed to the inherent difference in the local nature of typical (de)convolution and upsampling layers deployed in those networks, and the global nature of the long-term structural dependencies required by texture synthesis. In essence, texture synthesis involves distributing the input patch across the entire output grid while ensuring structural consistency. However, local (de)convolution layers usually operate in small scales, e.g. $3\times3$ or $5\times5$.

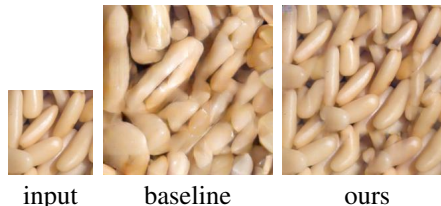

input&emsp;&emsp;baseline&emsp;&emsp;ours

Figure 1: Typical image-to-image translation CNNs, trained for synthesis, simply *upscales* the input, whereas our architecture naturally *expands* the input texture.

---

[*]Joint first authors, contributed equally.

As a result, they may simply enlarge the local content rather than naturally expand the input texture, even when trained for the texture synthesis task as seen in Fig. 1.

In order to develop a universal texture synthesis scheme, our unique perspective is to cast spatial texture synthesis as *local upsampling* in the Fast Fourier Transform (FFT) domain. Indeed, Fourier and spatial domains are *dual* of each other, where local characteristics in one translate to global characteristics in the other, and vice versa. This leads to an interesting property: for texture images with inherent repetitive patterns, that look (semi-)periodic, FFT of a small texture example is simply the downsampled FFT of the full-size texture. Rethinking texture synthesis as FFT upsampling, CNNs become a natural choice. FFT images however exhibit high dynamic range [21], and lack local correlations that impedes training of CNNs.

For effective training, we design a framework that performs FFT upsampling in the feature space. A convolutional encoder first extracts (multi-scale) features from input patch, which are then upsampled in FFT domain with a carefully designed deconvolution network, and subsequently decoded to construct large-size textures. Extensive evaluations are examined on various datasets with an array of quantitative and qualitative metrics. Our observations indicate both high generalization merit and fast speed that outperforms state-of-the-arts by large margins.

All in all, the major contributions of this paper are summarized as follows:

1. We, to the best of our knowledge, for the first time formulate the *global* task of texture synthesis as *local* FFT upsampling where CNNs endow generalization to unseen textures.

2. We design a novel framework for FFT upsampling in feature space to deal with high dynamic range and non-smooth nature of FFT images.

3. We perform extensive evaluations with an array of quantitative and human-based metrics that show the generalization merits of our scheme compared with several state-of-the-arts.

## 2 Related work

Texture synthesis has witnessed ample research. A holistic survey is beyond the scope of this paper; see e.g., [4]. Next, we list the most relevant works from the two main categories: non-parametric and parametric. Non-parametric examples include pixel-based [14, 58, 13], assembling-based [13, 39, 31, 49], optimization-based [48, 30, 51, 29], appearance-based [34], and image-analogy based [23] methods. Self-tuning texture optimization is state-of-the-art non-parametric method [29]. It matches certain global statistics (such as histogram) between input patch and output by optimizing a handcrafted objective. Thus, it can be prohibitively slow and brittle for complex textures.

Some recent parametric methods leverages deep CNNs to learn priors for data-driven synthesis. In particular, Gatys et al. [18] utilizes learned features from VGGNet to define a style loss that matches the grams between input and output. It requires iterative optimization, and thus can be prohibitively slow. Also, deep image analogies in [40] utilizes multi-scale features for data-driven synthesis. To speed up, feed-forward CNNs have been recently trained to learn the end-to-end synthesis map [55, 66, 52, 35, 36, 16, 27, 5, 3, 43]. Nevertheless, their underlying CNN architecture entails local operations with no component to capture the large-scale structural dependencies. As a result, one needs to train a CNN per each texture example that impedes generalization.

Texture synthesis is also closely related to style transfer [20, 37, 38], texture interpolation [62], inpainting [61, 60, 42, 65, 63, 50], and conditional image synthesis [26, 56, 46, 44, 11, 12]. The universal style-transfer [37] uses whiten-and-color transform (WCT) to transfer the style statistics to the content. Their primary goal is not texture synthesis, and when used for synthesis, it cannot preserve the structural fidelity required by texture synthesis. Also, the texture mixer method in [62] takes patches from different sources, and interpolates them by reshuffling and linearly mixing their latent codes in a controllable manner. However, the reshuffling cannot preserve the structural patterns of the input textures. Image inpainting and conditional image synthesis have also witnessed tremendous improvements using CNNs, but they typically deal with natural scenes and have not been designed for large holes.

Last but not least, FFT was previously used for texture synthesis but in a different way than our approach. In non-parametric methods, Galerne et al. [17] and Xia et al. [59] perturb FFT phase with random noise, while FFT spectrum is unchanged, to reproduce perceptually appealing micro-textures. In a different path, the parametric methods [54, 41] impose FFT spectrum constraints to effect low

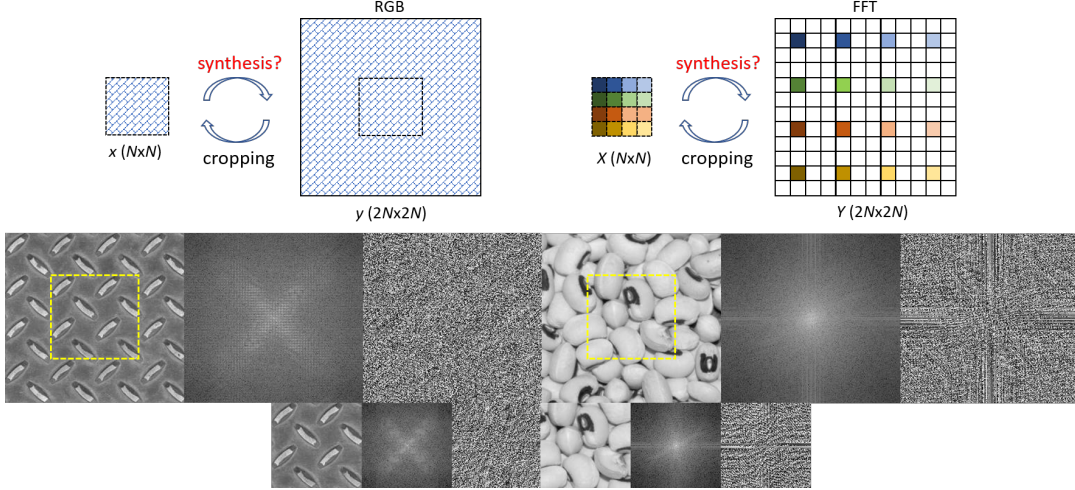

Figure 2: Schematic diagram (top): cropping in spatial domain (left) leads to downsampling in FFT domain (right). Real textures (bottom): gray-scale texture, FFT magnitude (log scale), and phase are shown for two examples and their center crops. FFT of full-size texture resembles upscaling the cropped feature FFT.

frequency structure. Tartavel et al. [54] promotes a variational-sparsity based synthesis, while Liu et al. [41] imposes FFT spectrum consistency for input and output on top of CNN-based feature matching in [19]. Our work, however, performs FFT in *feature* space to expand the features and synthesizes with a single network pass, which we find improves quality and speed significantly.

## 3 Preliminaries and problem statement

Consider a small image patch $x$ of size $N_1 \times N_2$. The synthesis aims to expand $x$ by a factor of $L_1 \times L_2$ to an $L_1 N_1 \times L_2 N_2$ image $y$ such that the visual features and structures are preserved both locally and globally. This is generally an ill-posed problem with possibly many valid solutions. A good solution hallucinates the structural patterns of the input in a seamless way which looks photo-realistic.

### 3.1 Texture FFT

To gain some intuition about the synthesis, it is useful to first look at repetitive textures in the FFT domain. Consider a simple 2D image $y(n_1, n_2)$ defined on the square grid of points $[LN] \times [LN]$, where $[LN] =: \{1, 2, \ldots, LN\}$. Suppose $y$ is repetition of a small $N \times N$ patch $x$ in a periodic manner as

$$y(n_1, n_2) = \sum_{l_1=0}^{L-1} \sum_{l_2=0}^{L-1} x(n_1 - l_1 N, n_2 - l_2 N) \tag{1}$$

Let $X$ denote the 2D FFT of $x$, that is defined as follows for $(k_1, k_2) \in [N] \times [N]$

$$X(k_1, k_2) = \frac{1}{N^2} \sum_{n_1=0}^{N-1} \sum_{n_2=0}^{N-1} x(n_1, n_2) \exp[-2\pi j(k_1 n_1/N + k_2 n_2/N)] \tag{2}$$

Similarly, $Y$ is FFT of $y$. Note, $X$ and $Y$ have the same size as $x$ and $y$, respectively. From the FFT definition (2), it can be shown that (see supplementary materials for the proof)

$$X(k_1, k_2) = Y(Lk_1, Lk_2), \qquad (k_1, k_2) \in [N] \times [N] \tag{3}$$

This simply means that in FFT domain, the patch $X$ is a downsampled version of $Y$; see Fig. 2 (top-right). Accordingly, as seen from Fig. 2 (top-left) starting from the small texture examplar $x$, one can perform upsampling to synthesize $y$.

Building on this intuition, our idea is to cast texture synthesis as upsampling in the FFT domain. However, realistic images contain aperiodic patterns, and textures may entail multiple structural

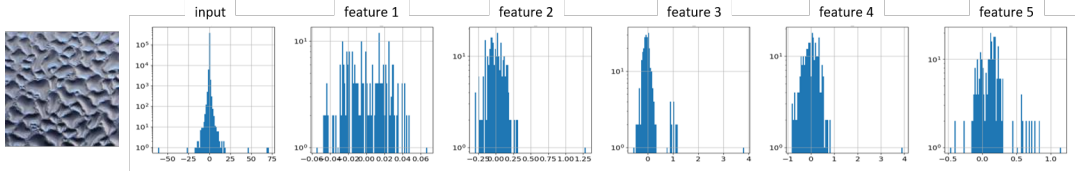

Figure 3: FFT Histogram for the input 128×128 texture (on the left) and five representative 16×16 feature maps randomly chosen from 512 feature maps. The dynamic range for input FFT is much higher than feature FFTs.

patterns with different orientations; see e.g., Fig. 2 (bottom). As a result, adopting traditional local interpolation techniques cannot produce seamless textures.

Training with data is therefore needed to learn an effective upsampler. For this means, we consider a supervised learning scenario where we have target images $\{y_i\}_{i=1}^{K}$ and from which we extract small patches $\{x_i\}_{i=1}^{K}$. Our goal is then, given the training data $\{(y_i, x_i)\}_{i=}^{K}$, learn the upsampler $h_\theta(\cdot)$ that maps $X$ to $Y$. Learning $h_\theta(\cdot)$ by effective design of neural networks is covered in the next section.

## 4 Neural FFT-based Texture Synthesis

CNNs have delivered tremendous success for RGB image-to-image translation tasks, and achieved state-of-the-art performance e.g., in single-image superresolution [28, 33]. One can naturally adopt those CNNs to learn the FFT upsampler $h_\theta(\cdot)$. The input RGB example is first converted to a FFT image, and CNN can be trained end-to-end to produce the FFT of the large texture. Training however becomes challenging because: 1) the FFT of natural images has high dynamic range (e.g., the zero frequency is noticeably larger that of other pixels), 2) FFT images are typically non-smooth and lack local correlations which is essential for the success of CNNs (see input image FFT in Fig. 3). In the following section, we discuss solutions for effective training in the FFT domain.

### 4.1 FFT upsampling in feature space

As an alternative to end-to-end training the synthesis map in FFT domain, we propose to use FFT upsampling in the feature space. The assumption is that feature maps are smooth with small dynamic range. To do so, we first use a convolutional encoder to extract $M$ feature maps, namely $\{h_i\}_{i=1}^{M} := f_{\theta_{enc}}(x)$ associated with the input image patch $x$ of size $N \times N$. Let us suppose the feature maps are of size $m \times m$ with $m \ll M$. In the experiments, we choose feature maps at three different scales $m = \{8, 16, 32\}$. Feature maps not only offer a smaller dynamic range, but also can decompose the existing texture into simpler patterns that are more amenable to upsampling and synthesis. As an example, compare the FFT histogram of input and representative features in Fig. 3.

Suppose we want to enlarge the texture by factor $L \times L$. The upsampling network operates independently on each feature map $h_i$. Let us define $\tilde{h}_i := \text{FFT}(h_i)$. The upsampler first converts feature map $h_i$ into two FFT channels $\{\text{Real}(\tilde{h}_i), \text{Im}(\tilde{h}_i)\}$ corresponding to real and imaginary components. These two channels are then fed into a pyramid deconvolution network $f_{\theta_{up}}$ to upsample features and generate $mL \times mL$ FFT feature $\tilde{H}_i$. In principle, we can imagine the $L \times L$ upsampler as a cascade of simple 2×2 upsamplers implemented each with a few learnable convolutional and transposed convolutional layers. The enlarged $\tilde{H}$ is then transformed back to RGB as $H = \text{IFFT}(\tilde{H})$. Note, $H$ is not necessarily real-valued, and we can impose conjugate symmetry on $\tilde{H}$ to ensure it is real-valued.

The upsampler is then followed by the convolutional decoder $f_{\theta_{dec}}(\{H_i\}_{i=1}^{I})$ that generates the large texture image. A typical choice is to adopt the U-net decoder with convolutional and unpooling (with nearest neighbor upsampling) layers that has been successful for image segmentation tasks. Similar to U-net, one can concatenate the features at multiple scales for more effective decoding. We perform concatenations for the outputs from upsampling 8×8, 16×16, and 32×32 feature maps.

### 4.2 Deformable convolution

The FFT upsampler elaborated before builds upon transposed convolutions to interpolate the FFT features. Transposed convolutional interpolation is suitable for uniform density upsampling, where the low resolution pixels are equi-distant on a regular 2D grid (e.g., as in Fig. 2). It can handle textures with (semi-)periodic patterns. However, for textures with non-periodic and irregular structural

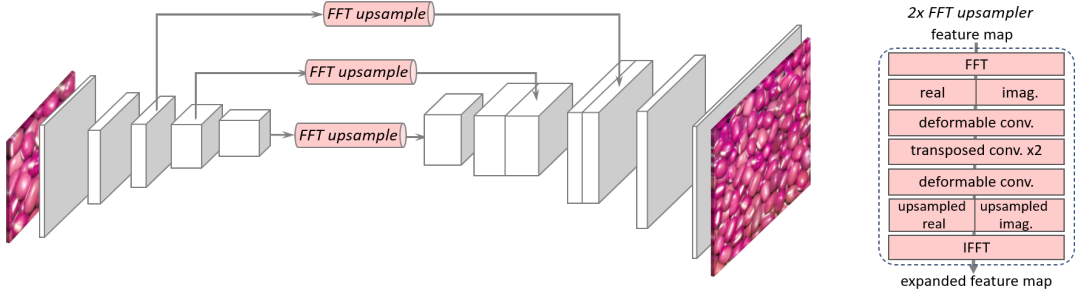

Figure 4: End-to-end network diagram for universal texture synthesizer. Convolutional encoder extracts features at multiple scales, upsampler with (parallel) deformable convolution network upscales FFT features by $2\times$, and the decoder concatenates the upscaled (multi-scale) features to generate the large texture.

patterns it can introduce tiling artifacts. Such scenarios demand an adaptive sampling strategy so as the interpolation kernel can arbitrarily select spatial samples from the entire FFT image. To do so, we leverage deformable convolutions [10] that can displace the pixels by learning an offset per each pixel. In essence, the offset re-arranges the pixel so as the relevant ones will fall into the kernel spatial window. With the designed architecture at hand, the following section discusses the training.

### 4.3 Training

The end-to-end synthesis network is trained using a mixture of perceptual, style, and adversarial loss. To promote perceptual quality, the VGG feature loss is adopted that is widely used e.g., in image super-resolution [28], synthesis [56], and inpainting [42]. Let $\Phi_p(x)$ with $h_p w_p c_p$ elements denote the concatenated features at layer $p$ of VGG-net for input $x$. We define the average perceptual loss as

$$\ell_{vgg} = \frac{1}{K} \sum_{i=1}^{K} \sum_{p=0}^{P-1} \frac{1}{h_p w_p c_p} \|\Phi_p(I_i) - \Phi_p(\hat{I}_i)\|_1 \tag{4}$$

which also acts as reconstruction loss. Moreover, we use style loss [19] to transfer structure. It imposes consistency between the gram matrices of features extracted from two images. Upon defining the $c_p \times c_p$ gram matrix $\Phi_p^\top \Phi_p$, the average style loss is defined as

$$\ell_{style} = \frac{1}{K} \sum_{i=1}^{K} \sum_{p=0}^{P-1} \frac{1}{c_p^2} \|\Phi_p^\top(I_i)\Phi_p(I_i) - \Phi_p^\top(\hat{I}_i)\Phi_p(\hat{I}_i)\|_1 \tag{5}$$

The perceptual and style loss require target images. To promote photo-realistic textures, the adversarial loss advocated in [22] is used as well. A patch-based discriminator, with parameters $\theta_d$, is trained. It takes the input patch $x$ and a few random patches from the generator output, namely $\hat{I} = f_{dec}(f_{up}(f_{enc}(x)))$. Let us collect all generator learnable parameters in $\theta_g := \{\theta_{enc}, \theta_{up}, \theta_{dec}\}$. Training then runs stochastic gradient descent to optimize the min-max objective

$$\min_{\theta_g} \max_{\theta_d} \lambda_{vgg}\ell_{vgg}(\theta_g) + \lambda_{style} \ell_{style}(\theta_g) + \lambda_{adv} \ell_{adv}(\theta_g, \theta_d) \tag{6}$$

## 5 Experiments

Performance of our FFT-based texture synthesis was assessed for a large and diverse dataset of natural texture images, and compared with state-of-the-art using both quantitative and qualitative metrics.

**Dataset**. A large texture dataset with $55,583$ images from $15$ different sources [8, 53, 9, 6, 7, 47, 1, 15, 45, 32] are collected. The dataset is quite diverse with a wide variety of patterns, scales, and resolutions. The dataset is randomly split into a training set of $49,583$ images, a validation set of $1,000$ images, and a test set of $5,000$ images. All images are resized with preserving the aspect ratio to the standard $256 \times 256$ size (or $512 \times 512$) for the target/output. $128 \times 128$ input patches are formed by cropping the center of target images.

## 5.1 Network architecture and training

The detailed architecture for 256×256 texture synthesis from 128×128 inputs is as follows.

**Encoder** starts with an RGB image patch 3×128×128, and after four pairs of stride=2 and stride=1 convolutional layers, it extracts 512×8×8. We take the features at 512×8×8, 256×16×16, and 128×32×32 for the FFT based upsampling.

**FFT upsampler** will first apply FFT on the encoded feature maps with each feature channel generating two components, real and imaginary, resulting in the size of 512×(2×8×8), 256×(2×16×16), and 128×(2×32×32). We will upsample the real and imaginary for each encoded feature channel separately. For instance, for each 2×8×8 complex feature, the upsampler is simply a shallow 5-layer network ordered as: 2 deformable convolutional layers, a transposed convolutional layer with stride=2, and 2 deformable convolutional layers. The output dimension is 2×16×16. Likewise, for the encoded feature maps' FFT components at 2×16×16 and 2×32×32 size. IFFT is applied per each channel's upsampled FFT components separately to arrive at 16×16, 32×32 and 64×64 feature outputs.

**Decoder** receives upscaled features at 3 scales namely 512×16×16, 256×32×32, and 128×64×64. It consists of 4 groups of nearest upsampling and convolution layers. Note, all layers expect the last one include batch normalization and ReLU. The output dimension is 3×256×256.

**Discriminator** takes input patch and 10 random 128×128 patches from the decoder. A convolutional layer first translates random crops to the feature space where they undergo 5 more convolutional layers with stride=2. Similar to [26] we employ the PatchGAN loss.

**Training** is implemented using Pytorch interface with cuDNN that has very efficient FFT modules. The model was trained on 4 DGX-1 stations with 32 total NVIDIA Tesla V100 GPUs and 320 CPUs using synchronized batch normalization layers [25]. We choose batch size of 8 per GPU, and the initial learning rate $10^{-5}$ that is halved every 200 epochs. Total of 800 epochs are used for convergence. We also set $\lambda_{vgg} = 0.1$, $\lambda_{style} = 200$, $\lambda_{adv} = 0.1$.

## 5.2 Quality metrics

We adopt a diverse array of metrics to assess the quality of generated textures. The quantitative ones compare the synthesized texture with the reference(s) using standard similarity metrics. A human evaluation is also included for a holistic comparison.

**Quantitative metrics**. We use Gram matrix score (GMS), learning perceptual image patch similarity (LPIPS) [64], FID [24], and structural similarity index metric (SSIM) [57]. GMS measures the image style fidelity as the normalized difference between gram matrix of output and targets based on (5). GMS and LPIPS require image pairs, but FID measures the distribution distance between the set of synthesized images and the ground-truth ones in the feature space. The crop-based version of these metrics are also used for textures, and thus termed c-GMD, c-LPIPS, and c-FID. c-FID computes the distance between 64 random crops of the output set and ground-truth set. c-GMS and c-LPIPS also average out the distance between the input patch and 8 random crops from the output.

**Human evaluation**. For a holistic comparison, we ask humans opinion. We use Amazon Mechanical Turk (AMT) to perform AB testing where the users are asked to choose between the synthesized textures from our method and one of the benchmarks, and provide a binary score. For each method pair, the orders are randomized, and 200 examples are viewed each by 15 users. A two-sample t-test is used to identify if the mean scores from two schemes is significantly different. The null hypothesis amounts to the means being equal, and it is rejected when the $p$-value is less than $10^{-6}$. Preference score (PS) indicates the portion of workers that prefer our result over the other method, averaged over 200 examples.

## 5.3 Results and comparisons

We compare against several leading benchmarks. Those include: 1) *naive tiling*: it simply duplicates the input 128×128 patch to form the 256×256 output; 2) *self-tuning* [29]: state-of-the-art optimization based method; two style transfer based methods including 3) *texture CNN* [18]: uses the 256×256 ground-truth as the input style and 256×256 noise as the content, and 4) *WCT* [37]: uses 128×128 patch as the style and 256×256 noise as the content; 5) *texture mixer* [62]: a texture interpolation method with all source patches chosen from the input; two GAN-based schemes 6) *sinGAN* [52] and 7) *nonstationary* [66] that overfits the network per each example; and 8) *pix2pix* [56]: an example for image-to-image translation methods. For all methods, we use the released codes by the authors.

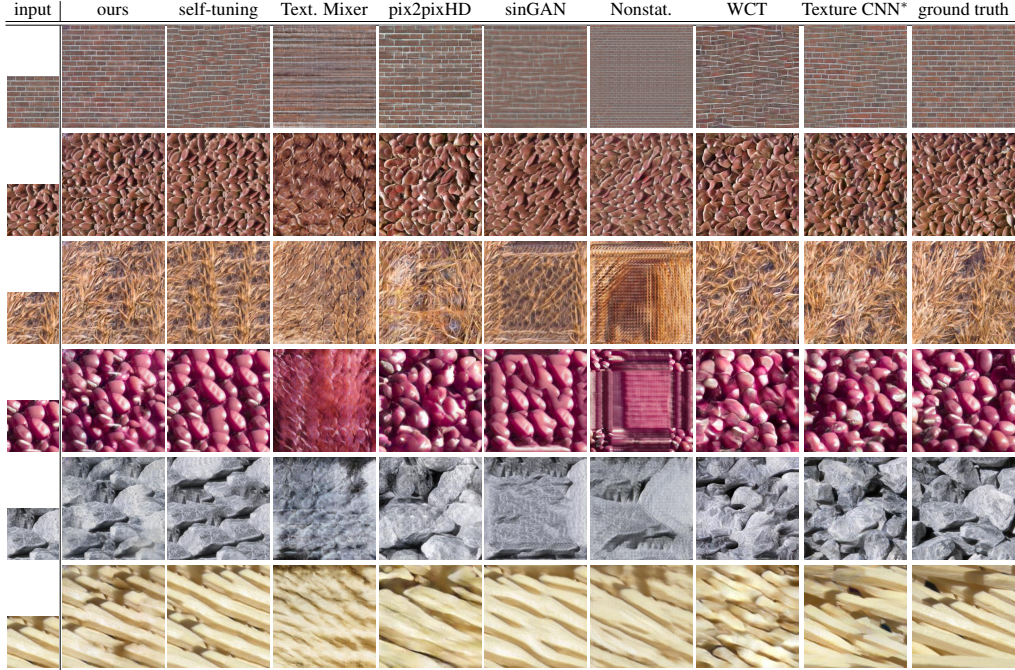

Figure 5: Results of different methods for $(128 \times 128) \rightarrow (256 \times 256)$ synthesis. Different from others, Texture CNN* uses the 256x256 ground-truth texture as the input style, and it is included for completeness. Ours achieves better synthesis than benchmarks on various types of unseen textures.

Table 1 lists the average quality metrics for our novel scheme on a test dataset of 200 images, and compares against benchmarks. The average preference score (PS), evaluated by 15 independent human readers, is also listed. We find that human readers prefer our method more often compared with all benchmarks. Our $p$-values for all comparisons are small ($\ll 10^{-6}$), indicating the preference of our method is statistically significant.

For other quantitative metrics, the results indicate that our method achieves the best scores except for GMS and FID. Texture CNN returns the best FID and GMS as it directly minimizes GMS. Note also that Texture CNN uses the full-size texture as the input rather than 128x128 input patch. Our scheme also significantly outperforms WCT, compare e.g., FID=128.1 vs. 71.82, which is mainly due to the structural artifacts present in WCT textures. This can be seen from the brick walls in Fig. 5 (first row). As evident by examples in Fig. 5, compared with Texture CNN and WCT, our method faithfully preserves the structures. With regards to speed, Texture CNN needs iterative optimization, and WCT demands expensive SVD computations. As a result both of them are slow; 45 ms for ours vs. 13 min for Texture CNN and 7 s for WCT.

The non-learning based self-tuning method is the closest to ours based on user preferences (PS=0.58). The synthesis looks realistic as seen from Fig. 5. However, the listed metrics (e.g., SSIM, c-FID, LPIPS) indicate that small structures are not well preserved; see the bricks in row 1, and beans in row 4. In addition, self-tuning tends to produce repetitive outputs. It is also more than $3 \times 10^3$ times slower than ours due to its iterative nature. Notice that self-tuning is not amenable to parallelization due to its iterative nature and the book-keeping mechanism. The source code we adopt also has been customized and implemented for CPU implementation. To have a fair comparison on CPU, neural FFT on CPU gives the run time of $1.29$ sec that is still $108\times$ faster than self-tuning method. Ours also significantly outperforms sinGAN and nonstationary methods (evident from PS scores 0.94 and 0.76, and FID=71.82 vs. 166.02 and 227.54). From Fig. 5, sinGAN tends to deform regular objects. Fig. 5 also shows that the pix2pix network tends to upscale the input patch; see e.g., rows 2,4.

For $4\times$ synthesis, we use the trained model for $2\times$ synthesis twice. Starting from the $128\times128$ input patch, we first generate $256\times256$ texture, and then use it as input to generate a $512\times512$ texture. The representative images are shown in Fig. 6. Further examples and evaluations for $2\times$ and $4\times$ synthesis, including scenarios that our approach fails, are provided in the supplementary materials.

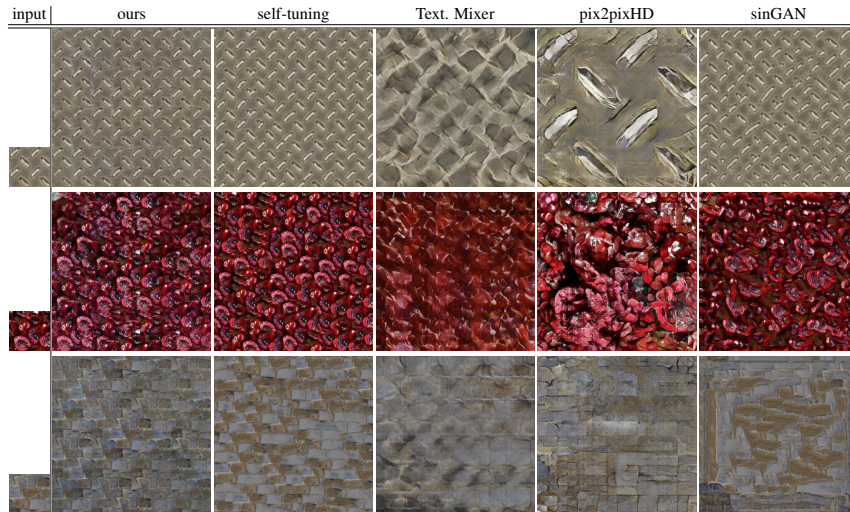

| input | ours | self-tuning | Text. Mixer | pix2pixHD | sinGAN |

Figure 6: Results of different methods on $(128 \times 128) \rightarrow (512 \times 512)$ texture synthesis. Ours produces better texture with higher fidelity and quality than benchmarks.

Table 1: Performance of different synthesis methods and ablation study for critical design elements averaged over 200 test examples. PS is based on scores received from 15 readers. Texture CNN* needs the ground truth as input. All methods are run on a single NVIDIA Tesla V100, except for self-tuning which runs the default 8 threads in parallel on an Intel Corei7-6800K CPU @ 3.40GHz. Note that the running time of ours is based on custom-ed CUDA kernel for deformable convolution.

|  | GMS | c-GMS | SSIM | FID | c-FID | LPIPS | c-LPIPS | PS % | time |
|---|---|---|---|---|---|---|---|---|---|
| Naive tiling | 0.0010 | **0.0011** | 0.28 | 79.22 | 0.59 | 0.35 | 0.28 | - | - |
| Self-tuning [2] | 0.0013 | 0.0016 | 0.29 | 99.45 | 0.40 | 0.36 | 0.29 | 58 | 140 s |
| Nonstationary [66] | 0.0026 | 0.0031 | 0.33 | 227.5 | 1.04 | 0.47 | 0.40 | 94 | 6 h |
| sinGAN [52] | 0.0016 | 0.0021 | 0.30 | 166.0 | 0.48 | 0.37 | 0.29 | 76 | 45 min |
| pix2pix [56] | 0.0011 | 0.0017 | 0.29 | 95.28 | 0.33 | 0.35 | 0.29 | 63 | 11 ms |
| Texture mixer [62] | 0.0022 | 0.0028 | 0.31 | 191.8 | 0.75 | 0.39 | 0.33 | 87 | - |
| WCT[37] | 0.0015 | 0.0019 | 0.28 | 126.1 | 0.4 | 0.37 | 0.30 | 67 | 7 s |
| Texture CNN*[18] | **0.0008** | 0.0015 | 0.29 | **68.97** | 0.38 | 0.34 | 0.29 | 61 | 13 min |
| No FFT | 0.0015 | 0.0020 | 0.29 | 147.6 | 0.43 | 0.40 | 0.32 | - | - |
| End-to-end FFT | 0.0015 | 0.0020 | 0.28 | 126.3 | 0.59 | 0.36 | 0.29 | - | - |
| No deform conv | 0.0011 | 0.0013 | 0.46 | 76.00 | 0.28 | 0.27 | 0.28 | - | - |
| **Ours** | 0.0010 | 0.0012 | **0.48** | 71.82 | **0.26** | **0.25** | **0.26** | - | 45 ms |

**Ablation study**. To demonstrate the critical elements in our designed architecture, we perform an ablation study for three different scenarios: 1) No FFT: CNN operates on encoded features without FFT. 2) End-to-end FFT: Synthesis is entirely in FFT domain, where network takes FFT of input patch and generates FFT of enlarged texture, 3) No deform conv: Deformable convolutions are dropped from the upsampler. The results are presented in Table 1, where we find that each proposed module plays an important role to achieve successful texture synthesis. For a representative texture, synthesized images for the scenarios considered in the ablation study are illustrated in Fig. 7.

**Nonstationary textures**. Notice that while neural FFT produce seamless textures for examples with repetitive patterns, it cannot handle well non-stationary textures. A few examples are shown in Fig. 8 where our method fails to produce a seamless texture. These are indeed challenging examples for any texture synthesizer.

# 6 Conclusions

This paper puts forth a novel FFT-based CNN framework for universal texture synthesis. In order to effect global characteristics, required by texture synthesis, this work casts synthesis as local upsampling in FFT domain. Also, for effective CNN training with non-smooth FFT images, we design a framework that applies FFT upsampling in the feature space using a deconvolution network

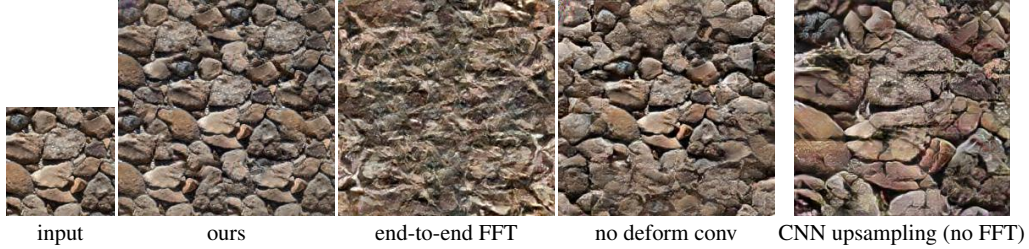

| input | ours | end-to-end FFT | no deform conv | CNN upsampling (no FFT) |

Figure 7: Illustration of ablation study for a representative example.

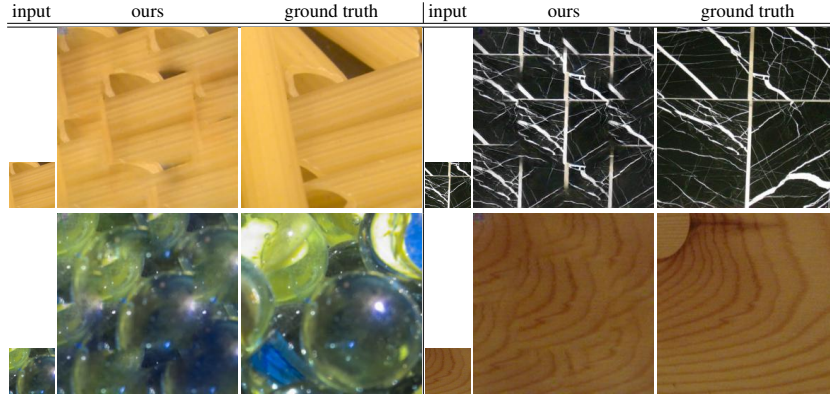

Figure 8: Failure examples for 2x synthesis. For texture with non-repetitive patterns and non-stationary objects our algorithm tends to tile.

with deformable conv layers. Our proposed scheme is generalizable and can synthesize arbitrary size textures from small inputs in real time. Extensive evaluations confirm that our synthesizer achieves state-of-the-art performance based on both quantitative metrics and human evaluations.

In order to address the shortcomings of the proposed approach, there are still important next steps to pursue for future research. One such step pertains to diversifying the generated texture in a controllable manner. Another step is to handle synthesis for non-stationary textures. As most non-stationary textures emphasize directional effects, one possible way to handle non-stationary textures could be emphasizing some specific FFT components while suppressing the others. Finally, the gram-based style loss may not effectively translate local characteristics, and thus developing a more effective criterion to match the input and output statistics becomes an important next step.

## 7 Broader Impact

Our AI research offers a powerful tool to synthesize a diverse range of textures with high fidelity and in a real-time manner. Our unique perspective of combining FFT, from signal processing tools, with deep learning for hallucinating images can be a great asset for other generation and style transfer tasks in graphics and vision. From the application standpoint, several applications in graphics and vision directly benefit from our tools to replace their tedious and manual synthesis platforms.

In particular, it helps rapidly create natural scenes for computer game developers, interior designers, and artists. In addition, our AI-based tool can discover the generation process behind the real-world scenes, which can help the professionals to better prototype ideas and create new textures.

In order to increase the positive impacts and reduce the downsides, we encourage further work to bring the users in the AI loop for additional guidance. This can allow artists to freely incorporate their creativity into the synthesis pipeline.

We also recommend the researchers and industries to investigate methods for further squeezing the CNN architecture, and efficiently implement them on the processing hardware. This would help not only make our tools faster for edge computing applications, but also reduce the high computational power consumed for training neural networks, that positively impacts the environment.

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
