[Supplementary Material]

# Supplementary Materials: Neural FFTs for Universal Texture Image Synthesis

**Morteza Mardani, Guilin Liu, Aysegul Dundar, Edward Liu, Andrew Tao,**
**Bryan Catanzaro** *
NVIDIA
{mmardani,adundar,guilinl,edliu,atao,bcatanzaro}@nvidia.com

In order to further evaluate our findings in the main paper, we provide additional supporting results and details. Unless otherwise stated, the evaluation setup mimics the setup adopted in the main paper. The list of contents is as follows:

- FFT derivations for texture synthesis (section 1)
- Further discussion about evaluation metrics (section 2)
- Additional evaluations and examples for 2x synthesis (section 3)
- Additional evaluations and examples for 4x synthesis (section 4)
- Examples of feature maps (section 5)
- Comparison with image quilting (section 6)
- Diversity (section 7)

## 1 Texture Syntheis as FFT Upsampling

### 1.1 FFT and IFFT definitions

For a 2D function $x(n_1, n_2)$ defined over the grid points $[N_1] \times [N_2]$, discrete Fourier transform (DFT) is defined as

$$X(k_1, k_2) := \frac{1}{N_1 N_2} \sum_{n_1=0}^{N_1-1} \sum_{n_2=0}^{N_2-1} x(n_1, n_2) \exp[-2\pi j(k_1 n_1/N + k_2 n_2/N)] \tag{1}$$

Note, fast Fourier transform (FFT) is an efficient implementation of DFT. To ease the exposition, throughout the main paper and the supplementary document, we refer to Fourier transform as FFT. Accordingly, Inverse FFT (IFFT) is also defined over the $[N_1] \times [N_2]$ grid as

$$x(n_1, n_2) := \frac{1}{N_1 N_2} \sum_{k_1=0}^{N_1-1} \sum_{k_2=0}^{N_2-1} X(k_1, k_2) \exp[2\pi j(k_1 n_1/N + k_2 n_2/N)] \tag{2}$$

### 1.2 FFT downsampling

In order to derive the relation $X(k_1, k_2) = Y(Lk_1, Lk_2)$ in equation (3) of the main paper, we begin with the FFT definition for $y$ as

$$Y(k_1, k_2) = \frac{1}{(LN)^2} \sum_{n_1=0}^{LN-1} \sum_{n_2=0}^{LN-1} y(n_1, n_2) \exp[-2\pi j(k_1 n_1/LN + k_2 n_2/LN)] \tag{3}$$

Recall the periodic function $y$ from equation (1) of the main paper as follows

$$y(n_1, n_2) = \sum_{l_1=0}^{L-1} \sum_{l_2=0}^{L-1} x(n_1 - l_1 N, n_2 - l_2 N) \tag{4}$$

Plugging (4) into (3), one arrives at

$$Y(k_1, k_2) = \frac{1}{(LN)^2} \sum_{n_1=0}^{LN-1} \sum_{n_2=0}^{LN-1} \sum_{l_1=0}^{L-1} \sum_{l_2=0}^{L-1} x(n_1 - lN, n_2 - lN) \exp[-2\pi j(k_1 n_1/LN + k_2 n_2/LN)]$$

After re-arranging the summation terms, and limiting their ranges to non-zero elements, we obtain

$$Y(k_1, k_2)$$
$$= \frac{1}{(LN)^2} \sum_{l_1=0}^{L-1} \sum_{l_2=0}^{L-1} \sum_{n_1=l_1 N}^{(l_1+1)N-1} \sum_{n_2=l_2 N}^{(l_2+1)N-1} x(n_1 - l_1 N, n_2 - l_2 N) \exp[-2\pi j(k_1 n_1/LN + k_2 n_2/LN)]$$

Now, defining the change of variable $n_1' := n_1 - l_1 N$ and $n_2' := n_2 - l_2 N$ yields

$$Y(k_1, k_2)$$
$$= \frac{1}{(LN)^2} \sum_{l_1=0}^{L-1} \sum_{l_2=0}^{L-1} \sum_{n_1'=0}^{N-1} \sum_{n_2'=0}^{N-1} x(n_1', n_2') \exp[-2\pi j(k_1 n_1'/LN + k_2 n_2'/LN + k_1 l_1/L + k_2 l_2/L)]$$

Since $\exp(2\pi j k) = 1$ for any integer $k$, it is easy to see that

$$Y(Lk_1, Lk_2) = \frac{1}{(LN)^2} \sum_{l_1=0}^{L-1} \sum_{l_2=0}^{L-1} \sum_{n_1'=0}^{N-1} \sum_{n_2'=0}^{N-1} x(n_1', n_2') \exp[-2\pi j(k_1 n_1'/N + k_2 n_2'/N)]$$
$$= \frac{1}{(LN)^2} \times L^2 \times N^2 \times X(k_1, k_2)$$
$$= X(k_1, k_2)$$

## 1.3  Additional examples for FFT upsampling

To further confirm the main idea in this paper about FFT upsampling, in addition to Fig. 1 (bottom) in the main paper, we provide additional examples below.

Figure 1: . Gray-scale image, DFT magnitude (in log domain), and FFT phase from left to right. The target and input images are shown on top and bottom parts.

## 2  Further discussion about evaluation metrics

There is no gold standard image quality metric, and most previous work on texture synthesis only provide visual comparison. As pointed out in section 5 of the main paper, we however provide a diverse array of eight metrics as each single metric could have its own bias. We included pair-wise similarity metrics such as SSIM and LPIPS, but it is apparent that they are not the best metrics for synthesis (where there are several good solutions), and thus we provided FID, c-FID, and c-LPIPS as well to compare the distribution of input/ground-truth images with the synthesized images. In order to compute the FID score, we measure the Frechet distance between the Inception-v3 statistics for a set of 200 synthesized images (resolution: 256x256) and the corresponding set of original (ground-truth, resolution: 256x256) images based on (3).

## 3  Additional 2× Synthesis Results

In this section we first report the quantitative quality metrics of our proposed neural FFT method for a a large test dataset of 5,000 examples, and compare with some state-of-the-art methods. We then illustrate representative texture synthesis outputs for our neural FFT method and compare with the exisitng methods.

### 3.1  Quantitative quality scores for 5,000 test dataset

Table I below lists the average quality metrics for 5,000 test examples. We compare with self-tuning, pix2pix, and WCT methods as representative state-of-the-art methods. We do not include other methods, reported in Table I of the main paper with 200 examples, as they are slow and collecting 5,000 outputs becomes quite time consuming.

Table 1: Performance of different methods for 2x synthesis averaged over a test set with 5,000 examples.

|  | GMS | c-GMS | SSIM | FID | c-FID | LPIPS | c-LPIPS |
|---|---|---|---|---|---|---|---|
| Self-tuning | 0.0012 | 0.0015 | 0.308 | 32.55 | 0.514 | 0.36 | 0.29 |
| pix2pix | 0.00095 | 0.00154 | 0.32 | 26.71 | 0.578 | 0.338 | 0.2732 |
| WCT | 0.00145 | 0.0019 | 0.302 | 63.1 | 0.468 | 0.3655 | 0.2915 |
| **Ours** | **0.00094** | **0.0012** | **0.495** | **20.74** | **0.2449** | **0.2586** | **0.26** |

### 3.2  Synthesized images

In addition to Fig. 5 of the main paper for $128 \times 128 \rightarrow 256 \times 256$ synthesis, a few more texture images are depicted in Fig. 2 to confirm the subjective quality of our neural FFT based synthesis.

**A note about Texture CNN**. It takes the ground truth as the input in the code provided by the authors. It cannot directly take the small 128x128 texture patch as the input. In order to make a fair comparison with our method we examined two scenarions: scenario 1) we replicated the input to create a 256x256 input image as the style; scenario 2) we upsampled the input to reach the 256x256 resolution. However, in both cases texture CNN fails to synthesize reasonable quality textures. For scenario 1, the synthesized texture looks tiled. For scenario 2, the synthesized texture also becomes upscaled. Therefore, we present the results for the texture CNN scheme where the ground truth as the input, and the synthesizer aims to reproduce a different texture image of the same size.

## 4  Additional 4× Synthesis Results

### 4.1  synthesized textures

In addition to Fig. 6 of the main paper for $128 \times 128 \rightarrow 512 \times 512$ synthesis, a few more texture images are depicted in Fig. 3 to confirm the subjective quality of our neural FFT based synthesis.

## 5  Feature maps

As stated in the main paper, we perform FFT upsampling in the feature space as it tends to exhibit a lower dynamic range, and be smoother. To further validate this idea, for a representative input 128x128 texture, the feature maps are depicted in Fig. 4. The bottom figure shows 256 encoded feature maps of size $16\times16$. For each 16x16 feature map the FFT magnitude and phase are also shown. The top row also shows the histogram of FFT coefficients (real and imaginary) parts for a few representative 16x16 feature maps.

## 6  Comparison with image quilting

For the sake of completeness we compare with image quilting (1) as one of the early pioneering methods for texture synthesis. Using the publicly available code [2], for two representative examples, the synthesis results are shown in Fig. 5. As evident from Fig. 5, image quilting performs poorly in synthesizing large-scale structures and multi-scale texture details. Similar observations have been made by prior works e.g., in (4).

## 7  Diversity

Notice that the primary focus of this work is to maintain structural fidelity. Producing a diverse collection of outputs from a single input, despite its importance, is not the central goal. Inspired by random phase noise models (5; 2), one simple approach is to perturb the FFT phase of feature maps (at different scales) prior to upsampling. Using variational training, one can then randomize phase in a controllable manner to generate a variety of output textures. This needs a systematic study that we leave for our future research.

## Footnotes

*The first two authors equally contributed.

[2]https://github.com/rohitrango/Image-Quilting-for-Texture-Synthesis

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

Figure 2: Results of different approaches on $128 \times 128$ to $256 \times 256$ texture synthesis. Texture CNN* takes the exact-size ground truth for test images.

Figure 3: Results of different approaches on $(128 \times 128) \rightarrow (512 \times 512)$ texture synthesis.

Figure 4: . Top: input texture and the histogram for some representative feature maps. Bottom: 256 feature maps of size 16x16 along with their FFT magnitude and phase. FFT of feature maps looks smooth.

Figure 5: Representative examples for neural FFT versus image quilting.