[Reviews · NeurIPS 2020]

Review 1

Summary and Contributions: Problem and Challenges: The paper presents Neural FFTs for universal texture image synthesis. Given a small patch of texture (128x128), the proposed approach allows us to synthesize texture on a larger area (256x256 and 512x512). This is a challenging problem: (1) only a small patch is available that makes it challenging to synthesize a wide variety of texture patterns; (2) the texture should be globally coherent; and (3) the texture synthesizer should be able to generate the texture in real-time. Contributions: (1) The authors present a very refreshing perspective of Fourier Transform for the problem of texture synthesis. The paper very nicely describes how to think about the problem from a frequency perspective and how we can use neural networks with these insights. (2) The texture synthesized using this approach is impressive and can be achieved in few milli-seconds.

Strengths: =============================== Why is this work exciting? =============================== This work is exciting because of the following reasons: (1) I always thought that texture synthesis is a local task but this paper gave me a fresh perspective about the global nature of this task. As we begin to appreciate this idea, the notion of using FFTs started becoming more natural. The authors very nicely present the idea in Section 3 and supplementary material. (2) The paper nicely introduces Neural FFTs with proper insights and motivations. (3) The results are impressive on the different qualitative examples shown in the paper and supplementary material!

Weaknesses: =============================== What is missing in this work? =============================== Here are the following things that I think are missing from this work and should be addressed: 1. Universal Texture Synthesis: The paper claims universal texture synthesis. However, it has been demonstrated to work regular texture patterns alone. There is a large variety of non-stationary texture (Zhou et al. [61]) that I think this work cannot address because of the fundamental regularity assumption or repetitive or stationary texture. 2. Competitive Baselines: I carefully looked through the outputs of Self-Tuning [2] and the results are equally impressive. The quantitative analysis and the human studies also seemingly suggest that. Impressively, [2] runs on a CPU with 8 core and the proposed formulation requires a Tesla V100 GPU. I would also point to the quality of results synthesized using Texture CNN. One may, however, complain about the amount of time it takes to synthesize a new texture using this approach. 3. I have some reservation about the evaluations. Please see the next section for specific details.

Correctness: The claims and method is technically correct. However, I think empirical analysis is missing some aspects. =============================== What is missing in the evaluation? =============================== 1. The authors use SSIM, FID, LPIPS along with AB testing. Here are my concerns: (a) SSIM, LPIPS contrasts an original image to the synthesized texture. There is no reason why a better SSIM and lower LPIPS would yield a better texture. This is evident when contrasting self-tuning [2] and proposed formulation. The results of both the approaches are equally good (also verified by the AB testing -- 58% users prefer proposed approach over [2]). (b) How is FID score computed? What is the resolution of images? What is considered to be the original data distribution? (c) Human Studies (CLARIFICATION) -- From Line 219, it sounds like 15 users viewed all 200 examples. I guess the authors want to say that - "they tested on 200 examples and each example is viewed by 15 random users"? 2. Apples-to-Apples comparison of computational time: Self-Tuning [2] impressively runs on a CPU with 8 core and the proposed formulation requires a Tesla V100 GPU. Training Neural FFTs also 32 v100 GPUs using a total of 4 DGX servers. Now getting self-tuning to work on a v100 gpu may not be trivial, can we do other way round? i.e. how much time would it take to run Neural FFT on the same resources as Self-Tuning [2]? 3. Comparisons with Image Quilting [Efros and Freeman, 2002]: Is it reasonable to ask the comparison with the texture synthesis from Efros and Freeman, 2002?

Clarity: Minor suggestions to improve the text: 1. Line 79-81: "Yu et al. [57]... However, .. input textures". This is texture mixer. How is it connected with the previous line? 2. Line 23-24: [34] is cited at both the places. I guess it should be one of the brackets, right?

Relation to Prior Work: I will make the following suggestions: 1. Line 63 should include citation to Image Analogies, Hertzmann et al., 2002. 2. Line 69-70 may include reference to Deep Image Analogies [Liao et al., 2018] that use multi-scale features for data-driven synthesis.

Reproducibility: Yes

Additional Feedback: Post-rebuttal comments: Dear Authors, Thanks for running the experiments for comparisons. The rebuttal is helpful in clarifying concerns and making the final decision. Here are my final suggestions to improve the draft: 1. The description in Line 4-10 of rebuttal should go somewhere in Sec-1. It is more helpful than Line 25-36 in current submission draft. 2. Inference time: This is a nice comparison with Self-Tuning [2]. It would be highly informative to the reader if the authors could include the discussion in main paper or appendix. 3. Thank you for running comparisons with Image Quilting. It would be great if it could be included in the final submission. 4. Evaluation Metrics: It would be good to add the discussion and details about the evaluation criterion in the appendix, especially human studies. E.g. Preference score is mostly A/B testing. It would be good to use the standard terms as well. 5. Figure-5 is overcrowded. It makes appreciation of the results of the proposed formulation slightly difficult. My suggestion would be to take-off some approaches and put them in the appendix. 6. Table-1 is overcrowded. My suggestion would be to keep Naive-Tiling, Self-Tuning and Texture-CNN for the comparison in main paper, and put the other comparisons/numbers in appendix. 7. We are on-board about Remark-1! It would be good to move it to appendix. Good work! --R1 ================================================ Pre-rebuttal comments: 1. Line 263 - 268: Remark 1 seems unnecessary and out-of-place. There is no reason why there has to be one good solution for texture synthesis. It is, therefore, I believe that evaluation criteria is wrong in this paper. (Note that I am not penalizing the proposed approach for not generating multiple plausible outputs). 2. Figure 5 (Supp. Material): It is for the above stated reason -- I don't see a good reason why the shown examples are failure cases if they do not match the "ground truth"?


Review 2

Summary and Contributions: The submission presents a novel method for texture synthesis using CNNs. Its main innovation is to treat texture synthesis as super-resolution in the fourier domain and successfully design a model architecture around that intuition that up-samples feature maps in fourier space. Update after rebuttal: I maintain my positive view on the submission.

Strengths: - Novel and original viewpoint on texture synthesis with CNNs. - Convincing results and decent evaluation.

Weaknesses: - Examples of model limitations should be shown in the main text (currently appendix Fig. 5)

Correctness: yes

Clarity: Overall yes, but the user study could be described more clearly. I am still not entirely sure what exactly the PS number means.

Relation to Prior Work: yes

Reproducibility: Yes

Additional Feedback:


Review 3

Summary and Contributions: This paper proposes to address the task of periodic 2D texture synthesis as Fourier space upsampling/superresolution with a deep network. The paper presents a straightfoward deep network architecture that first computes image features, takes the Fourier transform of these features, and uses deformable convolutions to learn the upsampling in Fourier space that corresponds to uncropping/texture synthesis. Comments after rebuttal/discussion: I stand by my original opinion that this paper should be accepted.

Strengths: The overall method presented in the paper is simple, straightforward, logical, and it seems to produce quite nice state-of-the-art results. The synthesized textures appear qualitatively superior to the presented baselines, and quantitatively superior in most metrics. I appreciate the authors' efforts to provide ablations that validate their design choices, as well as the use of many different quantitative quality/error metrics.

Weaknesses: In my opinion, the main weaknesses of the approach are that it seems predisposed to stationary texture synthesis (which should be easier than non-stationary), and that the presented results only show modest amounts of upsampling/synthesis. Many applications of texture synthesis (such as texturing graphics assets) would require much higher resolution output textures to be usable.

Correctness: Yes, I believe that the claims, method, and experiments are sound and correct.

Clarity: Yes, the paper is generally well-written and clear.

Relation to Prior Work: Yes, although I am not an expert in the field of texture synthesis, I believe that this paper adequately discusses prior work.

Reproducibility: Yes

Additional Feedback: I recommend editing the text to use fewer parantheticals (ex. the second sentence of the abstract), it can be a bit distracting for readers. line 17: replace comma with space line 25: that lack --> and cannot


Review 4

Summary and Contributions: This paper casts the image synthesis of repetitive texture patterns as local upsampling in the FFT domain. It performs neural FFT upsampling in multi-scale feature spaces to alleviate the difficulty of upsampling FFT signals in raw FFT domain, ie, high dynamic ranges and limited local correlations. The whole framework universally synthesizes arbitrary textures in a single forward pass. Qualitative evaluations show that the synthesized textures achieve the visually plausible performance and the quantitative results demonstrate the state-of-the-art quantitative performance on a list of metrics.

Strengths: - Treating repetitive texture synthesis as local upsampling in FFT domain is technically sound. I am glad that the authors provided informative illustrations and comprehensive mathematical derivations to prove the reliability of the proposed framework. - It is also an elegant and universal solution to this task. It does not parametrically model texture patterns in feature space (such as WCT), or learn texture patterns for individual texture image (such as SinGAN, Pix2Pix and etc.), or uses GT high-resolution texture image as the reference (such as TextureCNN). These solutions are known less desirable to comprehensively represent repetitive textural patterns and to synthesize them with global structural coherence. The proposed solution instead learns a simpler but universal local interpolation in feature-level FFT domain, and circumvents explicit modeling of the texture patterns, making the whole framework effective to arbitrary repetitive textures.

Weaknesses: Overall, I enjoy reading this paper: simple but effective idea, clear presentation, and convincing qualitative and quantitative results. However, I have certain concerns as follows. Please address the concerns in the rebuttal and incorporate the feedback in the final version. - The proposed method seems in principle not effective to non-stationary textures, as well as non-repetitive texture patterns. The conclusion discusses this issue a little bit and the supplementary materials showed some examples. I would like more discussions about the shortcomings, for example a short section to illustrate the failure cases and tell possible improvements upon the current framework that can handle them. - The ablation study only gave quantitative comparisons. But as it is a synthesis work, I expect qualitative results as well to convince the readers that each proposed module is effective and solves the limitations of the baseline. - The proposed upsampling operation uses a transposed convolution layer. I wonder whether the transposed convs can be replaced by other upsampling modules? For example, nearest neighbor upsampling + regular convolutional layer and etc. I am afraid a simple transposed conv will introduce aliasing artifact.

Correctness: Yes. The claims and method are correct.

Clarity: Yes. This paper is well written with comprehensive demonstrations, derivations and discussions. This paper is also well organized.

Relation to Prior Work: Yes. It has clearly discussed the differences or the improvements over the previous works.

Reproducibility: Yes

Additional Feedback:

[Author Response · NeurIPS 2020]

We would like to thank the reviewers for their positive and constructive comments, and for finding the idea novel, and
the results impressive. The major concerns are addressed below. The final paper will be updated accordingly. Also, the
typos will be fixed, and additional references suggested by the reviewers will be cited.
**Universal texture synthesis claim and non-stationary textures.** We agree that the proposed method does not handle
well non-stationary textures, and suits better textures with repetitive patterns. It, however, is universal in a sense that it
generalizes to unseen textures. Note, state-of-the-art methods such as [Zhou et al'18, Shaham et al'19] train and test
on a single texture and do not generalize to unseen examples. Having said that, looking at Fig. 5 (paper) and Fig. 2
(appendix), Neural-FFT handles a good range of regular, near-regular, and even irregular and stochastic textures. In
order to clarify this issue, as suggested by R4, we will add a discussion in the final paper, and move appendix's Fig. 5
(failure cases with non-stationary textures) to the main paper as suggested by R2.
**R1-1. Inference time comparison with self-tuning.** Self-tuning [Kaspar et al'15] is not amenable to parallelization
due to its iterative optimization nature and bookkeeping mechanism for the spatial uniformity constraint. Neural-FFT,
however, is quite parallelizable taking great advantage of GPUs. Having said that, we ran Neural-FFT in CPU mode
(same setting as self-tuning), and the inference time is: 1.29 sec versus 140 sec for self-tuning showing 108x faster
synthesis. Note, thanks to the parallel nature of Neural-FFT, inference on GPU takes 45 msec that is 3,111x faster than
self-tuning. We will include this discussion in the final paper. Regarding the synthesis quality, as shown in Fig. 2, and
Fig. 2,3 (appendix), self-tuning tends to produce repetitive outputs, and can break the regularities, whereas texture CNN
uses ground-truth textures for synthesis.
**R1-2. Comparison with [Efros and Freeman 2002].** Texture synthesis has advanced a lot after this pioneering work.
As per reviewer's suggestions, we compare with image quilting (using publicly available code[1]), and representative
examples are shown in Fig. 1. As evident from Fig. 1, image quilting performs poorly in synthesizing large-scale
structures and multi-scale texture details. Similar observations have been made by prior works e.g., in [Kaspar et al'15].

**R1-3. Evaluation metrics.** There is no gold-
standard image quality metric, and most previous
work on texture synthesis only provide visual com-
parison. We however provide a diverse array of
eight metrics as each single metric could have its
own bias. We agree that SSIM and LPIPS are not
the best metrics for synthesis (where there are sev-
eral good solutions), and thus we provided FID,

input        ours        image quilting        ours        image quilting

Figure 1: *Representative examples for neural FFT versus image quilting.*

c-FID, and c-LPIPS as well to compare the distribution of input/ground-truth images with the synthesized images. In
order to compute the FID score, we measure the Frechet distance between the Inception-v3 statistics for a set of 200
synthesized images (resolution: 256x256) and the corresponding set of original (ground-truth, resolution: 256x256)
images based on [Heusel et al'17].
**R1-4. Remark 1 and failure examples.** It seems that there is a misunderstanding. Indeed, remark 1 means that there
could be several good solutions for synthesis given an input example, where one can use a knob to control the trade-off
between structural similarity and diversity. The examples in Fig. 5 (appendix) are failures since based on the very first
definition of texture synthesis, the structural patterns are broken irrespective of the ground truth.
**R2-1. Clarifications on the PS number.** Preference score (PS) indicates the portion of workers that prefer our result
over the other method, averaged over 200 examples.
**R3-1. High resolution texture synthesis.** Regard-
ing the higher amounts of upsampling, a simple
heuristic is to take the trained 2x synthesis model,
and perform synthesis sequentially a few times
(fully convolutional network is invariant to the input
image dimension) to reach the desired resolution.
For example, for 4x synthesis results in Fig. 6 (main

input        ours        end-to-end FFT        no deform conv        CNN upsampling (no FFT)

Figure 2: *Illustration of ablation study for a representative example.*

paper) and Fig. 3 (appendix), we first perform 128→256 synthesis, and then 256→512 synthesis.
**R4-1. Qualitative results for the ablation study.** Thank you for the suggestion. A representative example is shown in
Fig. 1, and more examples will be added to the final version.
**R4-2. More discussions about the shortcomings** We discussed the shortcoming of no diverse output and possible
solutions in Sec. 5 of the paper. As most non-stationary textures emphasize directional effects, one possible way to
handle non-stationary textures could be emphasizing some specific FFT components while suppressing the others.
**R4-3. Alternatives to transposed convolution upsampling.** We have tested several other local upsampling methods
for FFT upsampling such as nearest neighbor, bilinear and trilinear interpolation. We empirically found that transposed
convolution works the best. We will include the results of these alternative upsampling experiments to the ablation
study.

## Footnotes

[1]https://github.com/rohitrango/Image-Quilting-for-Texture-Synthesis


[Meta-Review · NeurIPS 2020]

All four expert reviewers agree on the merit of the submission. The authors are encouraged to incorporate the suggestions of the reviewers into the final version.